# Ecosystem Services Trade-Offs and Synergies following Vegetation Restoration on the Loess Plateau of China

**Shutong Yang [1], Peng Shi [1,2,*], Peng Li [1,2], Zhanbin Li [1,2], Hongbo Niu [3], Pengju Zu [4] and Lingzhou Cui [5]**

1   State Key Laboratory of Eco-Hydraulics in Northwest Arid Region of China, Xi'an University of Technology, Xi'an 710048, China
2   Key Laboratory of National Forestry Administration on Ecological Hydrology and Disaster Prevention in Arid Regions, Xi'an 710048, China
3   Shaanxi Coalbed Methane Development Co., Ltd., Xi'an 710119, China
4   Shaanxi Ecological Industry Co., Ltd., Xi'an 710000, China
5   College of Life and Environmental Science, Wenzhou University, Wenzhou 325035, China
*   Correspondence: shipeng015@163.com

**Abstract:** The Loess Plateau (LP) is a heavily soil-eroded area. Since the year 1999, the Chinese government has carried out the "Grain for Green Project (GGP)" that has focused on increasing the regional vegetation coverage. Understanding the temporal and spatial variation of ecosystem services and the synergy in the LP is important for prospective regional re-vegetation and watershed administration. Therefore, three typical watersheds in the LP were selected: Huangfuchuan, Dalihe, and Yanhe. The spatial and temporal changes in carbon storage (CS), soil conservation (SC), and water yield (WY) in the watersheds were analyzed by the InVEST model from 2000 to 2020. Correlation analysis and root mean square deviation (RMSD) were used to investigate and compare the trade-offs in different ecosystem services (ESs). The results showed that the ES in the Huangfuchuan, Dalihe, and Yanhe watersheds overall developed in a positive direction, and increased from north to south. CS and SC showed a positive correlation in the three watersheds; however, there were negative correlations between CS and WY and between SC and WY. From 2000 to 2020, the trade-offs among CS, SC, and WY in the study area were in the descending order of the Yanhe, Dalihe, and Huangfuchuan watersheds, while the comprehensive benefits were in the opposite order. The results provided an essential basis for the high-quality development and ecological environment preservation of the Yellow River basin.

**Keywords:** ecosystem service (ES); spatial and temporal variation; trade-off and synergy; the Loess Plateau (LP); vegetation restoration

## 1. Introduction

An ecosystem is a dynamically balanced system composed of biological communities and their living environment, which not only provides material supplies for human beings, but also provides many indirect values for human development [1,2]. Existing studies and practices have shown that it is impossible to replace the functions of natural ecosystems [3,4]. Therefore, maintaining a pristine cycle of natural ecosystems is essential to improving the survival and development of human beings [5]. At the same time, various ESs are not independent to others, and their internal elements interact with each other in a complex way, which is often manifested as a trade-off or a synergistic relationship [6].

Recently, the trade-offs and synergies of ES became an essential research topic [7–10]. Jia et al. [11] studied the balance of ES on the Loess Plateau (LP) in northern Shaanxi from 2000 to 2008, and assessed the benefits of the "Grain for Green Project" (GGP). The results showed that quantifying the interactions between ESs can improve regional management practices to ensure the sustainable use of natural resources. Feng et al. [12] quantified the soil erosion control, carbon sequestration, and soil moisture and their interaction in

the Ansai watershed, and concluded that vegetation coverage and type were the leading factors affecting the trade-off. Xu et al. [13] evaluated the changes in ESs before and after the implementation of the GGP in the Zhifanggou watershed of the Loess Hills and Gullies region, where the watershed vegetation has undergone rapid recovery after severe damage in recent decades, and found a stable ecological development in the study area, with a 44.2% increase in the total value of ecosystem services from 1995 to 2010. Dong et al. [14] investigated the provision and requirement of ESs on the LP, and showed that vegetation restoration had a positive impact on the local ecosystem as well as on the regulation and supply services. Jafarzadeh et al. [15] examined the current land-use allocation systems in the Zagros area of western Iran, by analyzing the trade-offs and synergistic relationships between water production, prevention of soil erosion, carbon sequestration and marketable products through 533 sample sites, and the results showed that 75% of the studied sample sites had synergistic effects, and the highest synergistic effects were water production and prevention of soil erosion. Zheng et al. [16] analyzed the contribution of changes in agricultural land use intensity and type to grain production (GP) and water purification (WP), and their trade-offs in the Dongting Lake basin. The results showed that under the same climatic conditions, areas where agricultural land use intensity was the dominant factor were twice as likely as areas where land use type was the dominant factor. Yuan et al. [17] evaluated the supply, demand and trade-offs of and for food supply, soil and water conservation and carbon sequestration in Changzhi. The results show that there are trade-offs between food production and other services, and that ES cold spots are mainly located in built-up areas of the city.

Since the large-scale conversion of the GGP in the LP in 1999, the vegetation coverage has changed substantially [18,19]. By 2015, a vegetation recovery of 88.20% of the LP had achieved significant results. At present, most research on ecological restoration of the LP is focused on the value of and difference in the ES, and its response to land use conversion and ecological compensation, among other relevant research [20,21], whereas little attention has been paid to whether the trade-offs and synergistic relationships of the ES are consistent across precipitation, vegetation restoration types, vegetation cover and topography conditions. Wang et al. [22] assessed CS, WY and SC and their drivers in the LP from 2000 to 2018, showing that all three services increased during the study period and that mean annual precipitation (MAP) was the main driver of WY, while NDVI (Normalized Difference Vegetation Index) and a slope had the strongest explanatory power for CS and SC. Across climatic zones, land use is the most critical factor influencing the ES. Zhang et al. [23] evaluated the ES supply and demand relationships and the factors influencing them in the Loess Plateau region. The results show that supply services are generally trade-offs, with the slope and construction land area ratio being the main factors influencing ES trade-offs.

The trade-off and synergy of the ES after vegetation restoration could indicate the rise of one ES and the decline of another (trade-off relationship between different ESs), or the increase or decline (synergy relationship between different ESs) of both due to different revegetation strategies. The analysis of trade-offs can supply a basis on land-use planning and vegetation restoration strategy; however, vegetation restoration in different regions has different impacts on the trade-offs of the ES [24]. Therefore, we selected three watersheds with different levels of vegetation recovery from north to south in the central Loess Plateau as the study area and evaluated the changes in carbon storage (CS), soil conservation (SC), and water yield (WY) with the influence of vegetation restoration in the study area through the integrated valuation of ecosystem services and the tradeoffs (InVEST) model by remote-sensing interpretation data with precipitation data from 2000 to 2020. At the same time, we explored the trade-offs in different ESs in the watersheds, considered the mutual constraints of ESs, and explored the influence of vegetation recovery on ESs, which can enable us to find an equilibrium among economic and environmental factors.

## 2. Materials and Methods

### 2.1. Study Area

The LP (100°52′–114°33′ E, 33°41′–41°16′ N) covers a total area of 646,000 km², with numerous gullies and broken terrain (Figure 1). It has a continental monsoon climate, and the annual mean precipitation varies between 100–800 mm from northwest to southeast. The surface of the LP is primarily covered by loess sediment with a thickness of 50–200 m. The loess is loose and soft with serious soil erosion, covering an area of 454,000 km². The land use is largely comprised of grassland, forest land, and cropland, which is an important dry farming area in China [25].

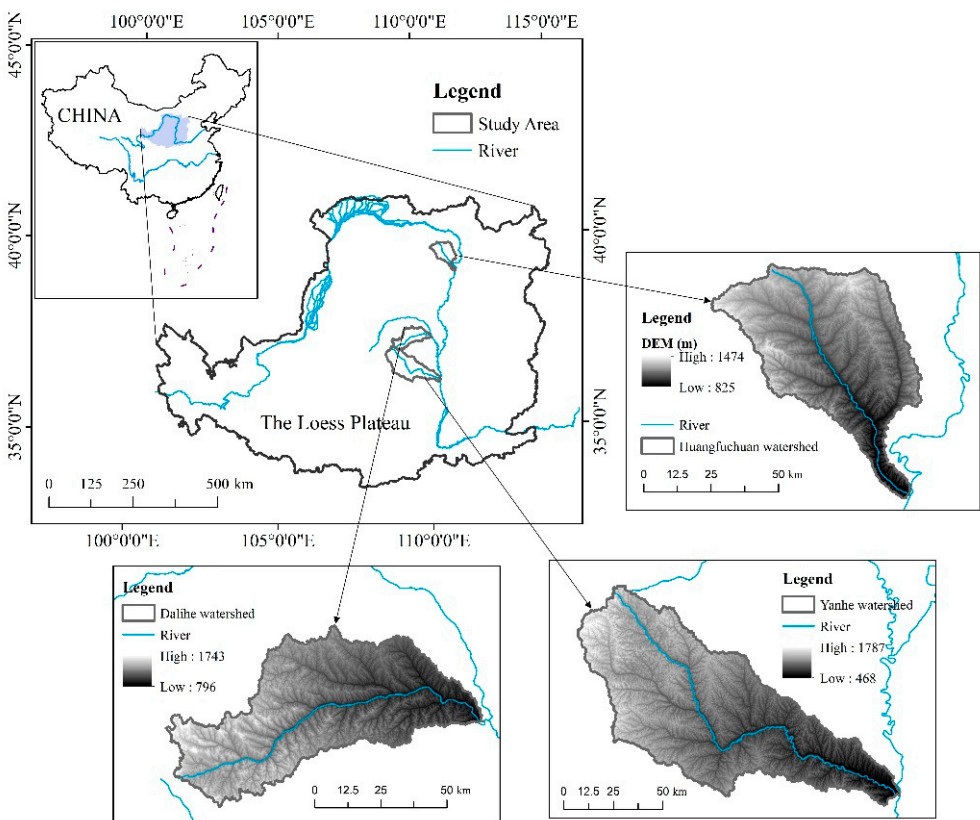

**Figure 1.** The location of study areas in the Loess Plateau, China.

The Huangfuchuan watershed (110°20′–111°15′ E, 39°12′–39°59′ N) is located in the hilly and gully area of southern Ordos, with an area of 3246 km², and the higher terrain in the northwest. The annual average precipitation is 395 mm. The annual average water surface evaporation is 1100 mm. Grassland is the main vegetation in the watershed. The water and soil loss in the watershed is serious and the terrain is broken. The Dalihe watershed (108°49′–110°14′ E, 37°36′–37°30′ N) is situated in the center of Wuding River, with an area of 3906 km². The average annual precipitation is 440 mm, with little inter-annual variation. The annual average water surface evaporation is 1500 mm. The vegetation in the watershed is mainly cropland and grassland. The Yanhe watershed (108°01′–110°27′ E, 36°27′–37°58′ N) is located in Yan'an City, with an area of 7687 km². The annual average temperature of the Yanhe watershed is 8.8–10.2 °C, and the annual average annual precipitation is 520 mm. Annual average water surface evaporation is 980 mm. The vegetation in the watershed is mainly grassland, followed by farmland and forest land.

### 2.2. ES Calculation

The DEM was derived from the Geospatial data Cloud (https://www.gscloud.cn/sources/accessdata/421?pid=302) (accessed on 10 August 2022). The coordinate system was UTM/WGS 84 and the spatial resolution was 30 m. The land use data was obtained from the 1:100,000 land use database in China, which can be downloaded from the Resource and Environment Science and Data Center (https://www.resdc.cn/data.aspx?DATAID=335) (accessed on 5 September 2022). The data are based on Landsat TM remote sensing image, and the interpretation accuracy is above 95%. The soil data were derived from the World Soil Database (HWSD). The dataset is provided by the National Cryosphere Desert Data Center (http://www.ncdc.ac.cn/portal/metadata/a948627d-4b71-4f68-b1b6-fe02e302af09) (accessed on 6 September 2022), and the coordinate system was WGS 84. The meteorological data were obtained from Shaanxi Meteorological Station, and the spatial distribution of precipitation was generated through the spatial interpolation tool of ArcGIS 10.2.

The InVEST model aims to evaluate the variations in the quality and value of ESs by simulating scenarios of different types of land cover, and to offer a scientific basis for the assessment and spatialization of ESs.

The enhancement of CS is of huge significance for reducing the concentration of carbon dioxide in the atmosphere and delaying global warming. The increase in SC can directly indicate that soil erosion has been effectively controlled. The SC module of the InVEST model quantifies the amount of SC and soil erosion in the ecosystem. Water availability determines to a certain extent the development potential of a region. The WY module of InVEST was used to obtain the WY based on the calculation of parameters such as precipitation, ground evaporation, and plant transpiration by using the principle of the water cycle (Table 1).

**Table 1.** Formulas of three ecosystem services.

| ES | Formulas | Explain |
|---|---|---|
| Carbon Storage (CS) | $C_{stored} = C_{above} + C_{below} + C_{dead} + C_{soil}$ | $C\_stored$ is the total carbon density in the study area (t·hm$^{-2}$), $C\_above$ is the carbon density in aboveground biomass (t·hm$^{-2}$), $C\_below$ is the carbon density in belowground biomass (t·hm$^{-2}$), $C\_dead$ is the carbon density in dead matter (t·hm$^{-2}$), and $C\_soil$ is the carbon density in soil (t·hm$^{-2}$). |
| Soil Conservation (SC) | $RKLS = R \times K \times L \times S$<br>$USLE = R \times K \times L \times S \times C \times P$<br>$A = RKLS - USLE$ | $RKLS$ is potential soil erosion (t·hm$^{-2}$·a$^{-1}$), $R$ is rainfall erosivity (MJ·mm·hm$^{-2}$·h$^{-1}$·a$^{-1}$), $K$ is soil erodibility (t·h·MJ$^{-1}$·mm$^{-1}$), $LS$ is a slope length-gradient factor (unitless), $USLE$ is actual soil erosion (t·hm$^{-2}$·a$^{-1}$), $C$ is a crop-management factor (unitless), $P$ is a support practice factor (unitless), and $A$ is soil conservation (t·hm$^{-2}$·a$^{-1}$). |
| Water yield (WY) | $Yield = \left(1 - \frac{AET}{P}\right) \times P$<br>$\frac{AET}{P} = 1 + \frac{PET}{P} - \left[1 + \left(\frac{PET}{P}\right)^{\omega}\right]^{\frac{1}{\omega}}$ | $Yield$ is annual water yield, $AET$ is annual actual evapotranspiration, $P$ is the annual precipitation, $PET$ is the potential evapotranspiration, and $\omega$ is a non-physical parameter that characterizes the natural climatic-soil properties. |

### 2.3. Analysis of Trade-Offs and Synergies

ESs do not exist independently, and correlation analysis can illustrate the orientation and magnitude of the interplay among services, and root mean square deviation (RMSD) can further indicate the trade-offs in ESs. This method can be used to quantify the mean

discrepancy within the standard deviation of a given service and the mean service standard deviation, and can indicate the degree of trade-offs between two or more ESs.

The formula used is as follows:

$$RMSD = \sqrt{\frac{1}{n-1} \times \sum_{i=1}^{n} (ES_i - \overline{ES})^2} \quad (1)$$

where $ES_i$ is the relative benefit value, $\overline{ES}$ is the average of ESs, and $RMSD$ is the distance from the spot to the arriswise. The RMSD was used to reveal the trade-offs of ESs between the Huangfuchuan, Dalihe, and Yanhe watersheds in 2000 and 2020.

Figure 2a shows the comprehensive benefits of the two ESs (normalized value), and the point on the dotted line in the figure indicates that the combined benefits of the two services are balanced. Figure 2b shows the trade-off between two ESs. The point on the dotted line indicates that the benefit values of the two services are equal, i.e., a zero trade-off.

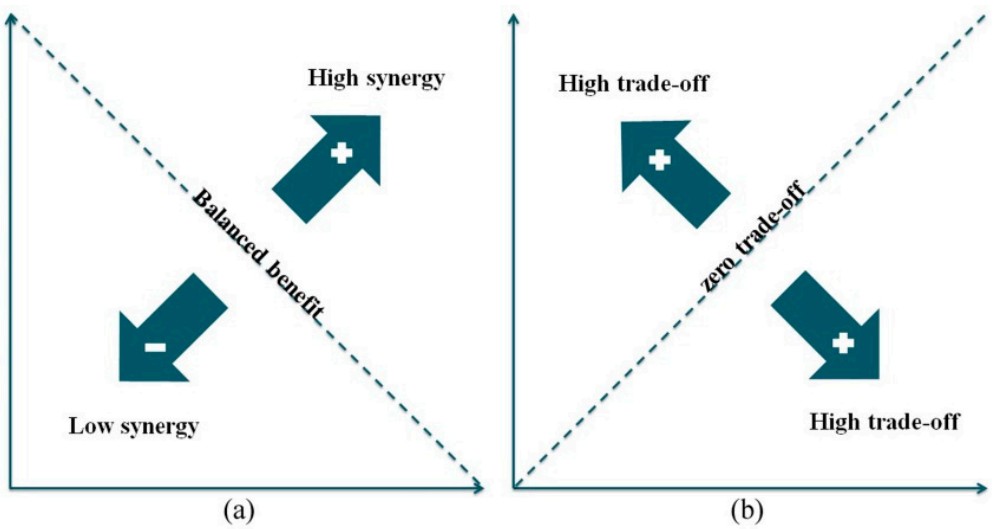

**Figure 2.** Figure depicting the trade-offs based on the standard deviation. (**a**) The comprehensive benefits of two ESs, (**b**) the trade-off between two ESs.

## 3. Results

### 3.1. Land-Use Changes

The main land uses in the Huangfuchuan watershed from 2000 to 2020 were grassland and cropland (Figure 3), which accounted for more than 85% of the watershed (Figure 4a). The cropland, water bodies, and urban land in the basin increased by 42.4, 2.6, and 31.4 km$^2$, respectively, the area of grassland decreased by 82.5 km$^2$, and the area of forestland remained essentially unchanged [26]. Over 90% of the basin area was cropland and grassland in the Dalihe watershed (Figure 4b). The areas of urban land increased by 6.1 km$^2$, and the area of cropland decreased by 3.6 km$^2$ from 2000 to 2020. The grassland and arable land in the Yanhe watershed occupied more than 80% of the area (Figure 4c). During the 20-year period, the areas of forest land, grassland, and urban land increased by 9.5, 9.0 and 72.7 km$^2$, respectively.

Since the various land use types had been converted mutually, the net changes in each land use type in the study areas from 2000 to 2020 was calculated (Figure 5). Although part of the cropland and grassland in the three watersheds had been transferred to forest land, the basic pattern that the cropland and grassland area dominated the various land use types had not changed. A part of the forest-land was transferred to other lands, but the forest-land had increased overall due to conversions from cropland, grassland, and other unused land. Over the past 20 years, the largest decrease in the Huangfuchuan watershed was grassland, and the largest increase in the area was cropland. The area of urban land

increased evidently in the Dalihe watershed, and the source was essentially from cropland and grassland. Much cropland land was transformed into forest- land, grassland and urban land in the Yanhe watershed.

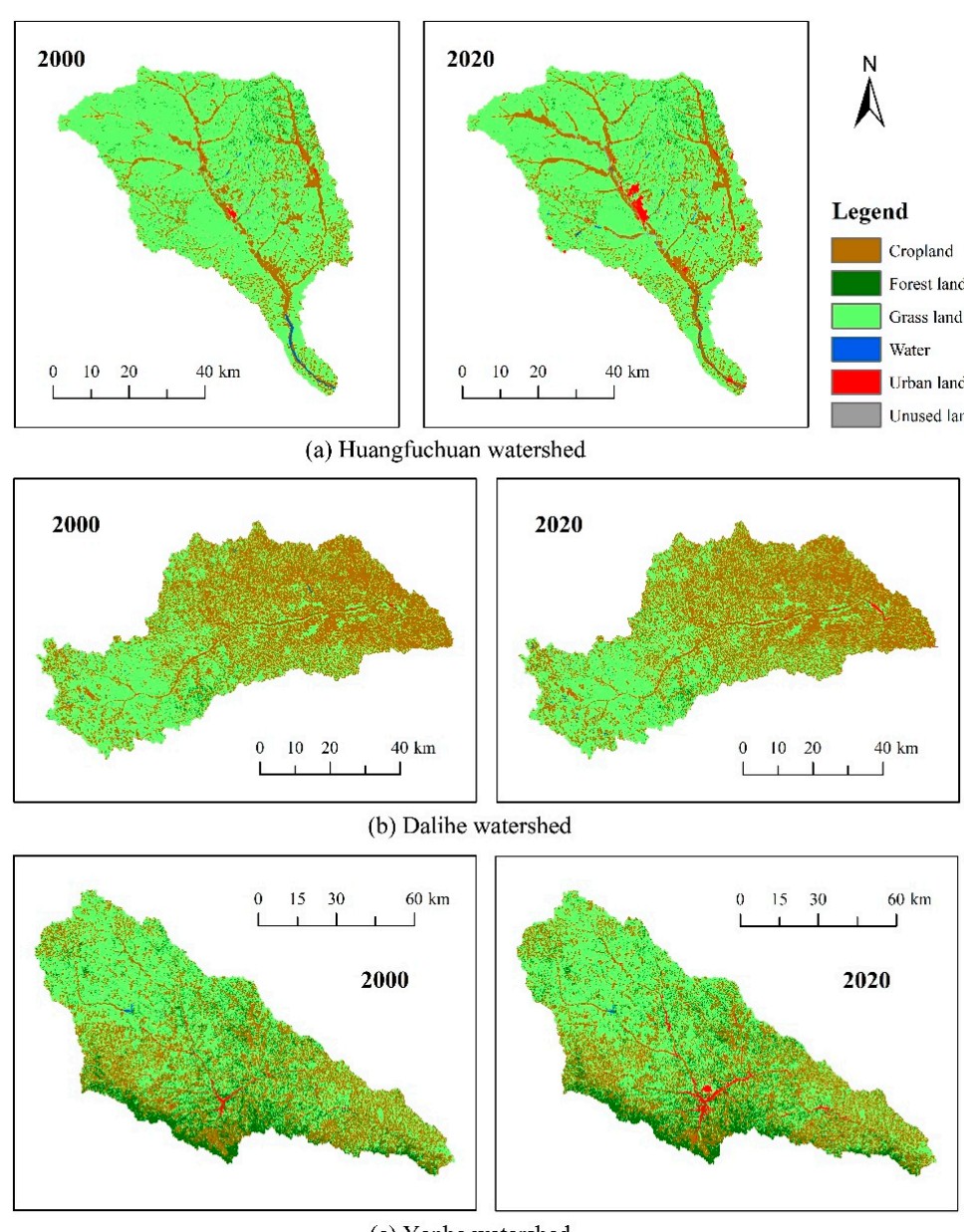

**Figure 3.** Land use in the study areas from 2000 to 2020.

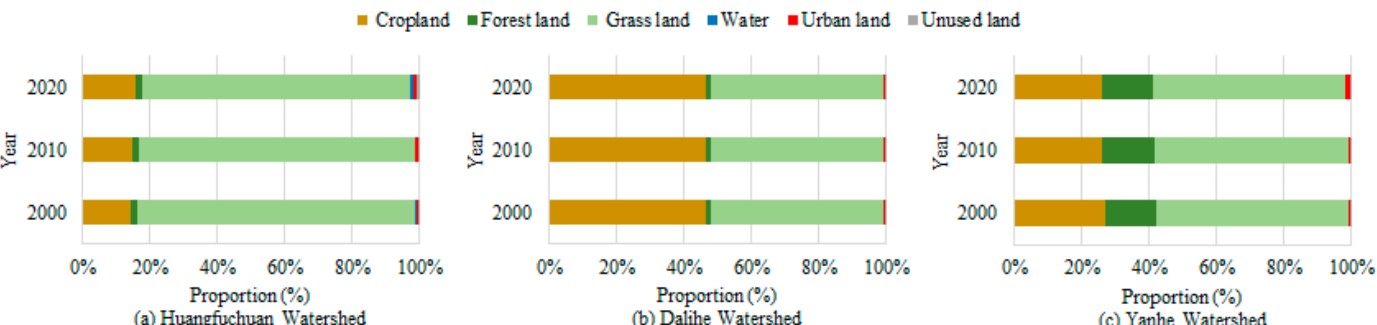

**Figure 4.** Proportion of different land use types in the study areas from 2000 to 2020.

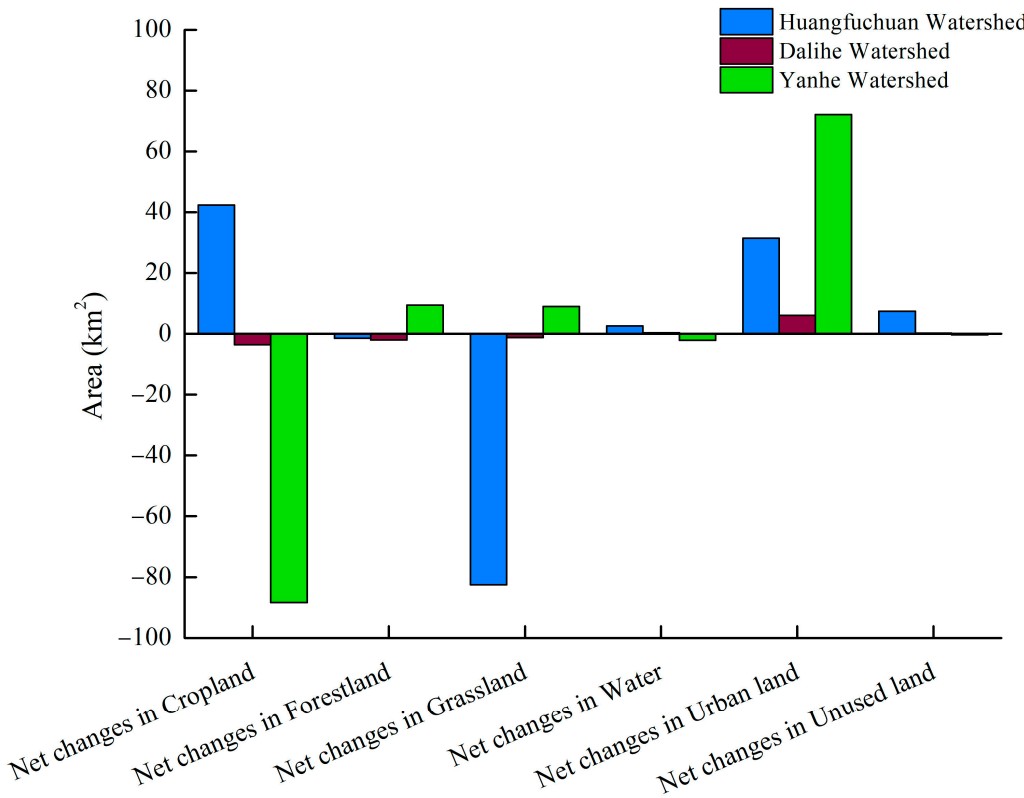

**Figure 5.** Net changes in each land use type in the study areas from 2000 to 2020.

*3.2. Spatio-Temporal Changes in ES*

3.2.1. Carbon Storage

The spatial dispersion of CS in the Huangfuchuan, Dalihe, and Yanhe watersheds during 2000–2020 was calculated by the InVEST model (Figure 6). The CS in the Huangfuchuan watershed showed a tendency to decline from 2000 to 2020. The CS moduli of the Huangfuchuan watershed from 2000 to 2020 were 5.13, 5.12, and 5.03 t·hm$^{-2}$, respectively. The total CS of the watershed in 2020 was 1.634 million t. From 2000 to 2020, the CS in the Dalihe watershed showed less change in the northern area of the watershed and greater change in the southern area. The CS moduli of the Dalihe watershed from 2000 to 2020 were 4.54, 4.53, and 4.52 t·hm$^{-2}$, respectively. The total CS of the watershed in 2020 was 1.767 million t. The CS in the Yanhe watershed kept stable during the 20 years. The southern area had a more evident change than the northern area. The CS moduli of the Yanhe watershed from 2000 to 2020 were 5.67, 5.68, and 5.65 t·hm$^{-2}$, respectively. The total CS of the watershed in 2020 was about 4.341 million t.

Comparing the CS in the study area, we see that the average CS in the Huangfuchuan, Dalihe, and Yanhe watersheds from 2000 to 2020 was between 5.03–5.13, 4.52–4.54, and 5.65–5.68 t·hm$^{-2}$, respectively. According to the distribution of CS in the study area, the CS modulus of the three watersheds was in the following descending order: Yanhe > Huangfuchuan > Dalihe. Combining Figures 3 and 4, we see that the Yanhe watershed has a greater proportion of grassland and forestland, the Huangfuchuan is predominantly grassland, while the Dalihe has more arable land, suggesting that forestland and grassland contribute more to CS in the watershed than arable land.

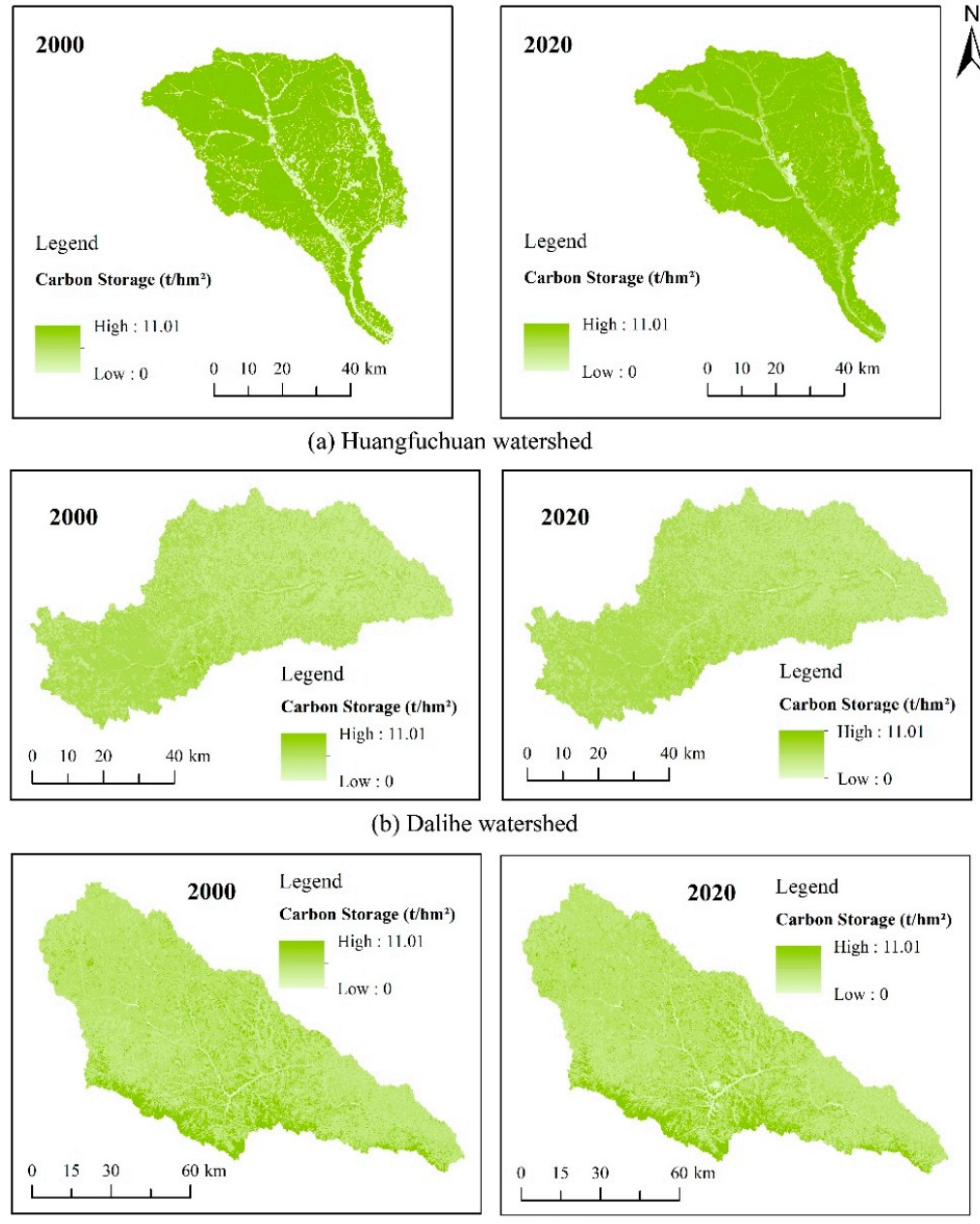

**Figure 6.** Distribution of carbon storage in the study areas from 2000 to 2020.

### 3.2.2. Soil Conservation

The spatial distribution of SC in the Huangfuchuan, Dalihe, and Yanhe watersheds during 2000 to 2020 was calculated by the InVEST model (Figure 7). From 2000 to 2020, the SC in the Huangfuchuan watershed showed an increasing trend, and the southern area showed a more evident change than the northern area. The SC moduli of the Huangfuchuan watershed from 2000 to 2020 were 28.94, 70.74, and 79.72 $t \cdot hm^{-2}$, respectively. The amount of SC in 2020 was 25.88 million t.

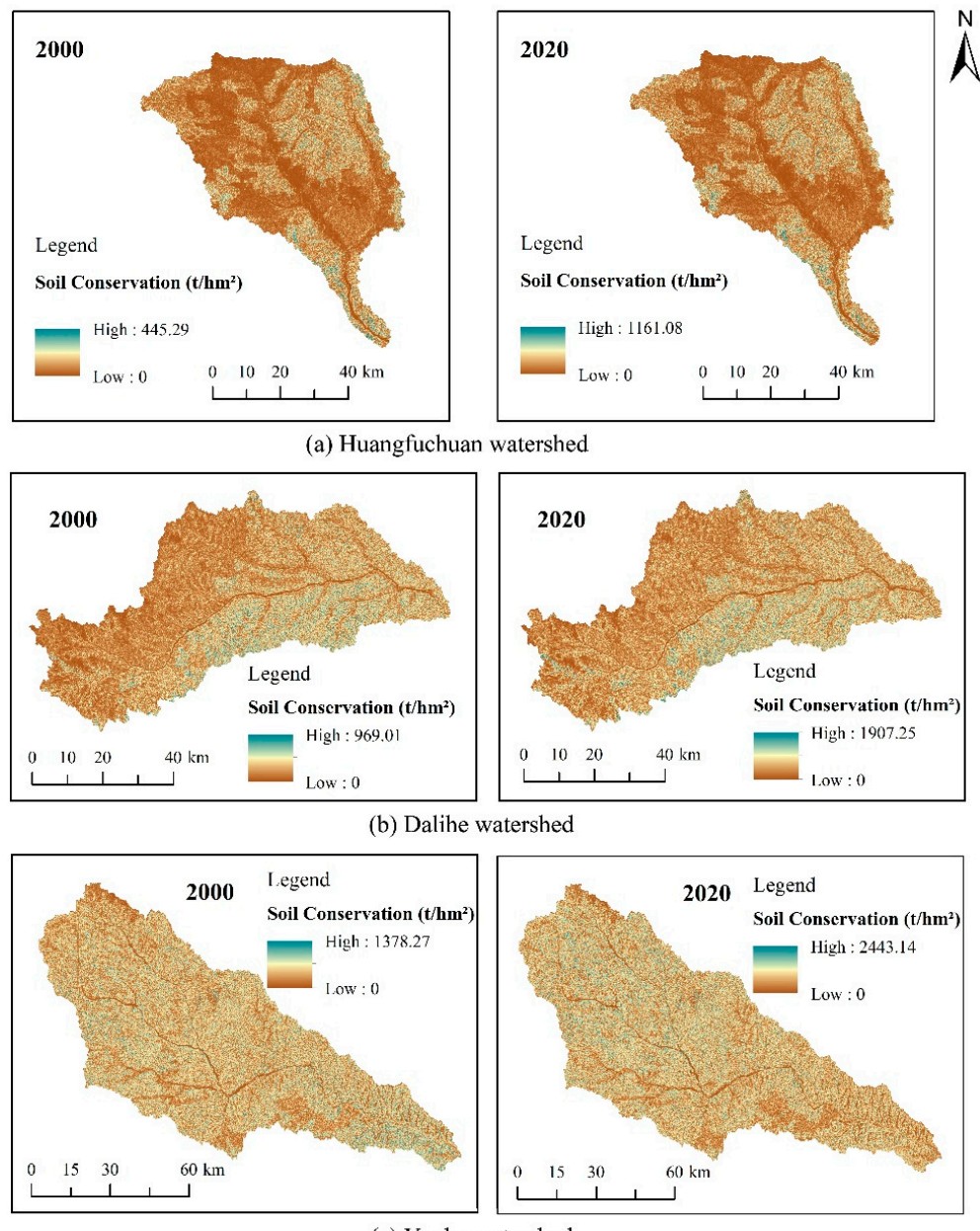

**Figure 7.** Distribution of soil conservation in the study areas from 2000 to 2020.

The SC in the Dalihe watershed showed an increasing trend during the 20 years, with marginal changes in the upstream area of the watershed and large changes in the downstream area. The SC moduli of the Dalihe watershed from 2000 to 2020 were 127.18, 174.41, and 247.72 t·hm$^{-2}$, respectively. The amount of SC in 2020 was approximately 96.76 million t.

The SC in the Yanhe watershed increased with marginal changes in the northern area of the watershed and large changes in the southern area. The SC moduli of the Yanhe watershed from 2000 to 2020 were 249.82, 349.91, and 434.08 t·hm$^{-2}$, respectively. The amount of SC in 2020 was approximately 333.67 million t.

If we compare the SC in the study area, we see that the average SC in the Huangfuchuan, Dalihe, and Yanhe watersheds from 2000 to 2020 was between 28.94–79.72, 127.18–247.72, and 249.82–434.08 t·hm$^{-2}$, respectively. According to the distribution of SC in the study area, the SC modulus of the three watersheds was in the following descending

order: Yanhe > Dalihe > Huangfuchuan. The increase in SC in the study area demonstrates the effective management of soil erosion, i.e., the ecological effect of the GGP is evident.

### 3.2.3. Water Yield

From 2000 to 2020, the WY in the Huangfuchuan watershed increased (Figure 8), and the southern area had a more evident change than the northern area. The WY moduli of the Huangfuchuan watershed from 2000 to 2020 were 5.67, 68.95, and 89.12 mm, respectively. The amount of WY in the watershed in 2020 was 289.30 million m$^3$.

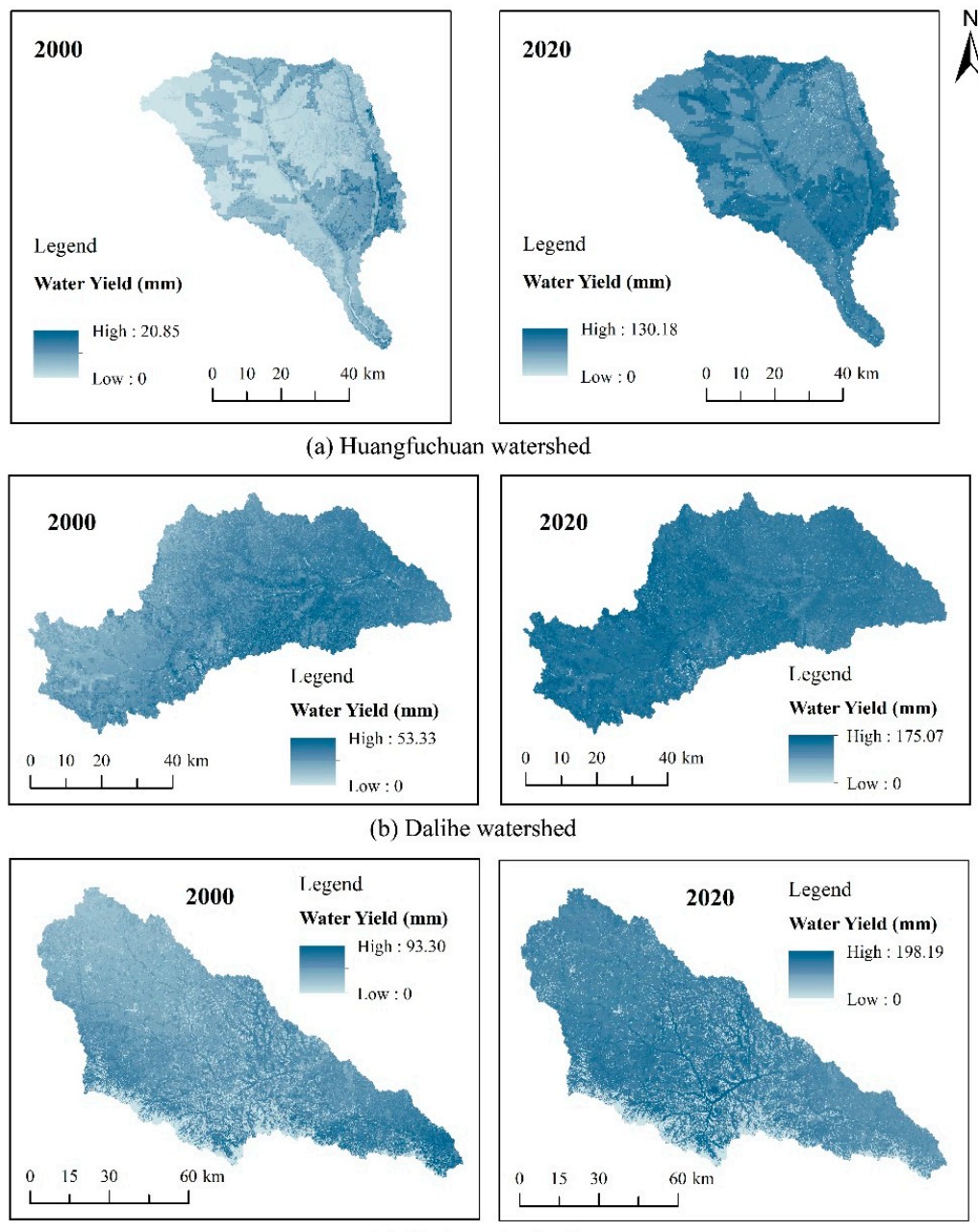

**Figure 8.** Distribution of water yield in the study areas from 2000 to 2020.

The WY is significantly affected by precipitation. The WY in the Dalihe watershed increased during the 20 years, with marginal changes in the central area of the watershed and large changes at the edge. The WY moduli of the Dalihe watershed from 2000 to 2020 were 30.23, 65.23, and 125.41 mm, respectively. The amount of WY in the watershed in 2020 was approximately 489.87 million m$^3$.

The WY in the Yanhe watershed increased, and the southern area had a more evident change than the northern area. The WY moduli of the Yanhe watershed from 2000 to 2020 were 39.23, 79.77, and 117.26 mm, respectively. The amount of WY in the watershed in 2020 was 901.41 million m$^3$.

Comparing the WY in the study area, we see that the average WYs of the Huang-fuchuan, Dalihe, and Yanhe watersheds from 2000 to 2020 were between 5.67–89.12, 30.23–125.41, and 39.23–117.26 mm, respectively. According to the distribution of WY in the study area, the WY modulus of the three watersheds was in the following descending order: Yanhe > Dalihe > Huangfuchuan. The distribution of WY is closely related to the amount of annual precipitation and evapotranspiration. Geographically speaking, the Huangfuchuan watershed is located in the north, where annual precipitation is low, and the low WY is in line with natural laws.

In general, the increase in forestland and grassland in the study area has had a positive impact on ESs, with CS, SC and WY all showing an increasing trend from 2000 to 2020.

*3.3. Relationships among Ecosystem Services*

In 2000, the correlations among the carbon storage, soil conservation, and water yield of the Huangfuchuan watershed were as follows (Table 2): CS vs. SC (R = 0.0887), CS vs. WY (R = −0.3643), and SC vs. WY (R = −0.1934). In 2020, the correlations of the three ES were as follows: CS vs. SC (R = 0.1554), CS vs. WY (R = −0.4726), and SC vs. WY (R = −0.2775).

**Table 2.** Correlation among ecosystem services from 2000 to 2020.

| Years | Watersheds | Ecosystem Services | Carbon Storage | Soil Conservation | Water Yield |
|---|---|---|---|---|---|
| 2000 | Huangfuchuan | Carbon Storage | 1 | | |
| | | Soil Conservation | 0.0887 ** | 1 | |
| | | Water Yield | −0.3643 ** | −0.1934 ** | 1 |
| | Dalihe | Carbon Storage | 1 | | |
| | | Soil Conservation | 0.0139 ** | 1 | |
| | | Water Yield | −0.6588 ** | 0.1892 ** | 1 |
| | Yanhe | Carbon Storage | 1 | | |
| | | Soil Conservation | 0.0154 ** | 1 | |
| | | Water Yield | −0.8471 ** | 0.0186 ** | 1 |
| 2010 | Huangfuchuan | Carbon Storage | 1 | | |
| | | Soil Conservation | 0.0852 ** | 1 | |
| | | Water Yield | −0.5451 ** | −0.1965 | 1 |
| | Dalihe | Carbon Storage | 1 | | |
| | | Soil Conservation | 0.0366 ** | 1 | |
| | | Water Yield | −0.7817 ** | −0.0629 ** | 1 |
| | Yanhe | Carbon Storage | 1 | | |
| | | Soil Conservation | 0.0216 ** | 1 | |
| | | Water Yield | −0.9161 ** | −0.0007 ** | 1 |
| 2020 | Huangfuchuan | Carbon Storage | 1 | | |
| | | Soil Conservation | 0.1554 ** | 1 | |
| | | Water Yield | −0.4726 ** | −0.2775 | 1 |
| | Dalihe | Carbon Storage | 1 | | |
| | | Soil Conservation | 0.0092 ** | 1 | |
| | | Water Yield | −0.6993 ** | −0.1032 ** | 1 |
| | Yanhe | Carbon Storage | 1 | | |
| | | Soil Conservation | 0.0190 ** | 1 | |
| | | Water Yield | −0.2339 ** | −0.0521 ** | 1 |

** indicate significant differences at the 0.01 level.

In 2000, the correlations among the carbon storage, soil conservation, and water yield of the Dalihe watershed were as follows: CS vs. SC (R = 0.0139), CS vs. WY (R = −0.6588),

and SC vs. WY (R = 0.1892). In 2020, the correlations of the three ESs were as follows: CS vs. SC (R = 0.0092), CS vs. WY (R = −0.6993), and SC vs. WY (R = −0.1032).

In 2000, the correlations among the carbon storage, soil conservation, and water yield of the Yanhe watershed were as follows: CS vs. SC (R = 0.0154), CS vs. WY (R = −0.8471), and SC vs. WY (R = 0.0186). In 2020, the correlations of the three ESs were as follows: CS vs. SC (R = 0.0190), CS vs. WY (R = −0.2339), and SC vs. WY (R = −0.0521).

Carbon storage and soil conservation showed a positive correlation in the three watersheds, i.e., the two ESs were in a synergistic relationship. However, there was a negative correlation in groups of carbon storage and water yield, and in soil conservation and water yield; namely, these two groups of services were in a trade-off relationship. From 2000 to 2020, the correlation between ESs in the Dalihe watershed decreased, but this correlation increased in the Huangfuchuan and Yanhe watersheds.

### 3.4. Trade-Offs and Synergies of ES

From 2000 to 2020, the trade-off between SC and WY in the Huangfuchuan watershed (Figure 9) increased, and the trade-off between CS and SC decreased. In the Dalihe watershed, the comprehensive benefit value between CS and WY increased, the degree of trade-off between SC and WY decreased, and the comprehensive benefit value increased. In the Yanhe watershed, the trade-off between SW and WY increased, and the overall benefit value increased. It can be concluded that the trade-off relationship among the three services of CS, SC, and WY in the three watersheds was in the following order: Yanhe < Dalihe < Huangfuchuan, while the relationship of comprehensive benefit value was in the following order: Yanhe > Dalihe > Huangfuchuan. From 2000 to 2020, the trade-off relationship between CS and WY in the study area decreased, while the comprehensive benefits increased.

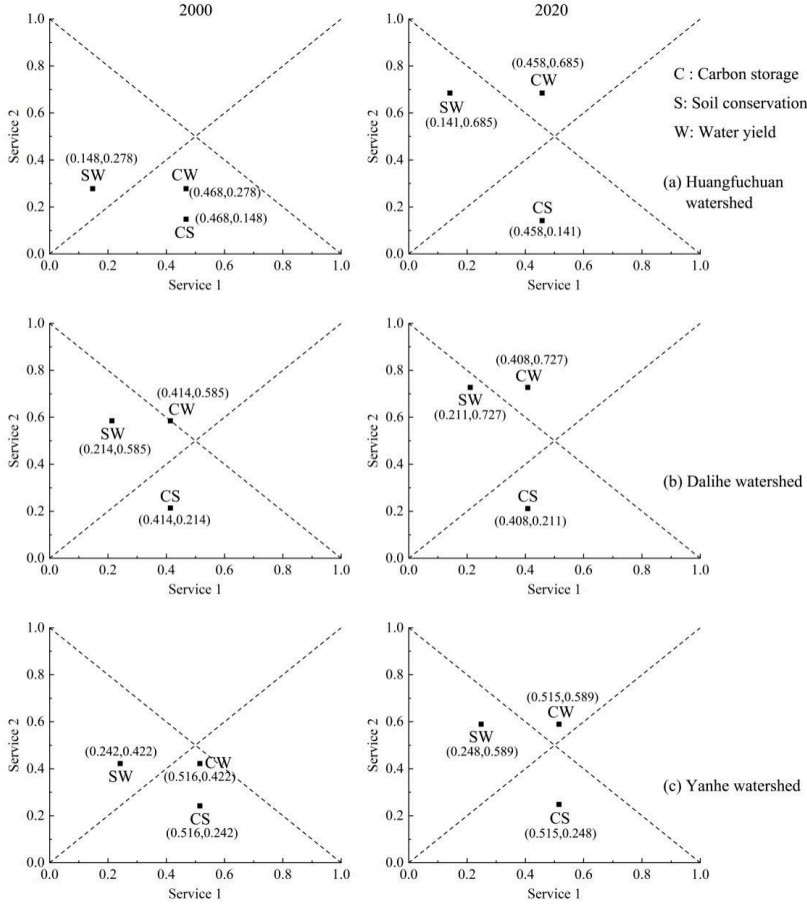

**Figure 9.** Trade-offs between ecosystem services in 2000 and 2020.

## 4. Discussion

### 4.1. Uncertainty of Data and Methods

Based on remote sensing data, this study evaluated the ecosystem services and their trade-offs in the study area, and the results are relatively reliable, but there are still some limitations. For example, different types and degrees of uncertainty may be introduced at various stages in the life cycle of remotely sensed data, from data acquisition, processing, and analysis, and propagated during subsequent processing. The study area is located in the central part of the Loess Plateau, which is characterized by long surface gullies and complex soil and environmental factors; consequently, uncertainties in the remote sensing data, including land use data, may have an impact on the results of the study [27,28]. Therefore, the source and nature of uncertainty in remotely sensed image data should be understood as much as possible in the next step of the study, and the impact of uncertainty should be reduced as much as possible [29].

At the same time, model assessment also has limitations. The InVEST model is an open-source evaluation model. The model simplifies the process of ecosystem services and cannot fully restore reality, relying more on improving the accuracy of parameter acquisition and parameter calibration to ensure the accuracy of the assessment results. As a result, uncertainty in the raw information data also leads to a certain amount of error in the analysis and calculation results, which has an impact on the accuracy of the assessment of carbon storage, soil conservation and water conservation in regional ecosystems and leads to some limitations in the analysis results.

Thirdly, the study is of limited value in ecosystem services selection. There are many kinds of ecosystem services. Based on the literature, this study selected three (carbon storage, soil conservation and water yield) that have a greater impact on the Loess Plateau, without considering the impact of other services (such as biodiversity), and the conclusions may not be comprehensive.

Finally, the scale of the study also has limitations. The three watersheds selected for this study cover an area of 3246, 3906 and 7687 km$^2$, and the assessment of ecosystem services and their trade-offs and synergies in the study area on this scale may have limited reference value for larger- and smaller-scale study areas. Future research could therefore be oriented towards different scales, looking at trade-offs and synergistic relationships between ecosystem services in more macro or micro domains.

### 4.2. Comparison with Previous Studies

Land use conversion is an important relevant topic in current research [30], which has a significant impact on climate change and ecosystems [31] and is directly related to the conservation and maintenance of ESs [32]. The results showed that vegetation restoration on the LP had positive effects on regional ESs. These outcomes are similar to previous studies [33,34]. Compared with other land uses, forestland and grassland can provide better ESs. Xie et al. [35] calculated the value of 11 types of ES in China, and showed that the total value of forest is the highest. Luo et al. [36] analyzed the transformation in land use, and ES supply and their interactions in typical small catchments on the LP of China over the past half-century, and concluded that the best possible ratio between grassland and woodland (approximately 1.5) may support higher levels of synergistic ecosystem services.

The ecosystem is a complex system; its services do not exist independently, and complex interactions exist among its internal elements. Therefore, the management process of ESs can be considered as a process of trade-off between various ESs to a certain extent [37]. Since the conversion under the GGP in 1999, the vegetation coverage of the LP has been evidently increasing, and the vegetation recovery potential is 69.75%. The areas with a high vegetation recovery potential index are concentrated in the northern sandstorm area and the western hilly and gully region [25]. In other words, there are differences in vegetation restoration in different regions of the LP, and their impacts on ESs also varied. In this study, a large amount of arable land in the Yanhe River basin was converted to forest- and

grassland, and its soil conservation and water yield increased rapidly while the degree of trade-offs decreased and the overall benefits increased; that is, the vegetation restoration on the LP has had a catalytic effect on regional ecosystem services and a positive impact on ES synergy [38,39]. It shows that different land use policies affect the function of regional ecosystems and can provide a theoretical basis for future vegetation restoration efforts.

*4.3. Suggestions on Management Policies*

More than 60% of the Huangfuchuan watershed is grassland. The high value of ESs was mainly in forestland and grassland, which played essential functional roles in the entire watershed. Therefore, the Huangfuchuan watershed can be considered as a site to improve the structure and the quality of forestland and grassland on the premise of satisfying the mandatory factors, policy-oriented factors, economic coordination factors, and ecological security factors. The cropland accounts for about 50% of the Dalihe watershed, and the integrated ES of the cropland was low. Therefore, promotion of regional soil and water protection can be continued according to local conditions, and the decrease in ESs caused by human activities can be minimized. The area of urban land in the Yanhe watershed has increased rapidly. It is the main site for human activities, and its ESs are low. In the future, without damaging the existing ecological environment, the focus can be on urban water, soil conservation, and the conservation and maintenance of forests and grasslands in watersheds. Simultaneously, the area of the forestland can be increased and its structure improved, and the vegetation coverage of the same type of land use can be increased [24,40].

In the study area from 2000 to 2020, the vegetation in the three watersheds developed in an overall positive direction. Therefore, areas with the highest comprehensive ES can be protected and changes in land use types can be restricted. In the second-highest value area, protection can be continued to restrict its construction and development, and its development toward higher services can be promoted. Low-service areas that are considerably affected by human activities can be combined in the future with the on-site environment, focusing on soil and water conservation according to local conditions, and controlling further reduction in its services. At the same time, ecological reserves can be scientifically designated to optimize the structure of land use, promote comprehensive land improvement, and enhance the quality of arable land, thereby improving the service capacity of regional ecosystems.

## 5. Conclusions

Based on a comparison of the assessment of the ESs, and the distribution of ESs within each watershed: The ESs in the Huangfuchuan, Dalihe, and Yanhe watersheds overall developed in a positive direction, and increased from north to south. CS and SC showed a positive correlation in the Huangfuchuan, Dalihe, and Yanhe watersheds, i.e., the two ES were in a synergistic relationship. However, there were negative correlations between CS and WC and between SC and WC, i.e., these two groups of services were in trade-off relationships. It can be concluded that the trade-offs in the CS, SC, and WY in the study area were in the following order: Yanhe Watershed < Dalihe Watershed < Huangfuchuan Watershed, while the comprehensive benefits were the opposite. From 2000 to 2020, the trade-off between CS and SC in the study area decreased, while the comprehensive benefits increased. These were closely related to the degree of vegetation restoration in the study area. The Grain for Green project has increased the capacity of carbon storage, oxygen release and climate regulation in the study area, and improved the support and regulation services of the ecosystem. However, there were still trade-offs between the different services. Future revegetation should therefore be based on field conditions and the selection of appropriate plants to avoid trade-offs between ecosystem services triggered by blind tree planting.

**Author Contributions:** S.Y., P.S., P.L. and Z.L. conceived the main idea of the paper. S.Y. and P.S. wrote the manuscript. H.N., P.Z. and L.C. contributed to improving the paper. All authors have read and agreed to the published version of the manuscript.

**Funding:** This work was supported by the Project of Creating Ordos National Sustainable Development Agenda Innovation Demonstration Zone (Grant 2022EEDSKJXM005), the National Natural Science Foundation of China (Grant 42077073), the Natural Science Basic Research Plan in Shaanxi Province of China (2022KJXX-62), and the Project of Shaanxi Coal and Chemical Industry Group Co., Ltd. (2022SMHKJ-A-J-07-02, 2022SMHKJ-B-J-54).

**Institutional Review Board Statement:** Not applicable.

**Informed Consent Statement:** Not applicable.

**Data Availability Statement:** Not applicable.

**Acknowledgments:** We thank the reviewers for their useful comments and suggestions.

**Conflicts of Interest:** The authors state that they have no known competing financial interest or personal relationships that could affect the work described in this article.

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
