# Peer review of "Ecosystem Services Trade-Offs and Synergies following Vegetation Restoration on the Loess Plateau of China"

_sustainability, doi:10.3390/su15010229_

Round 1

Reviewer 1 Report

The authors used InVEST model to analyze the spatial and temporal patterns and relationships of ecosystem services in LP in detail. I have a few major concerns that require serious revision by the author.

In the abstract and introduction, the paper only emphasizes the significance of practice and application, and does not point out the research progress and academic vacancy of ecosystem services in LP region. Please supplement and highlight the academic significance of this part.

The subgraphs of Figure 3,6,7,8 should be labeled (a) (b) (c)... , the legend and its position of all graphs remain unified.

The writing of the discussion section is not very clear.  It is suggested that the author sort out 2-3 sub-headings to discuss.  It should at least include the uncertainty of the method, the discussion of the main results of the paper with previous studies, the practice of management policy and the prospect of future research direction, etc.

The conclusion is not a simple summary of the results, but the re-emphasis of the research and academic significance of the full text, and the prospect of the core results of the article and the enlightenment of future research and practice.

Reviewer 2 Report

This study analyzed the spatial and temporal changes of carbon storage, soil conservation, and water yield in the watersheds of three typical watersheds in the Loess Plateau by the InVEST model from 2000 to 2020, and compared the trade-offs in different ecosystem services. The results of this study may contribute to the understanding of the temporal and spatial variation of ecosystem services and the synergy in the Loess Plateau. However, there are some concerns that the authors should address before it can be considered for publication.

(1) The authors should add more information (including the significance) about the study area and further explain why the Loess Plateau is selected as the study area.

(2) I suggest the authors add more information about the land use types of the three watersheds in "2.1. Study area".

(3) In the data, I suggest that authors add more information about data, such as data availability and access.

(4) I suggest the authors further discuss the impact of mutual transformation of different land use types on ecosystem functions.

(5) More mechanism explanations should be added to further explain the temporal and spatial changes of ecosystem service functions.

(6) A paragraph of limitation discussion should be added to clarify the limitation or uncertainty of data and methods in this current study. For example, the uncertainty of remote sensing data including land use data (Decuyper et al., 2020; Li et al., 2022; Shen et al., 2020, 2022) may affect the research results.

(7) The conclusion is not a simple restatement of the results. The authors should further clarify the contribution of the research results to the research field.

(8) I suggest the authors improve the resolution of Figure 1, and add watershed names in Figure 3, 6, 7, 8.

References:

Spatio-temporal assessment of beech growth in relation to climate extremes in Slovenia–An integrated approach using remote sensing and tree-ring data. Agricultural and Forest Meteorology, 2020, 287, 107925.

Uncertainty of city-based urban heat island intensity across 1112 global cities: Background reference and cloud coverage. Remote Sensing of Environment, 2022, 271: 112898.

Marshland loss warms local land surface temperature in China. Geophysical research letters, 2020, 47, e2020GL087648.

Asymmetric impacts of diurnal warming on vegetation carbon sequestration of marshes in the Qinghai Tibet Plateau. Global Biogeochemical Cycles, 2022, 36: e2022GB007396.

Reviewer 3 Report

The work done is very interesting, uses the software well and has important results. However, in general, the manuscript is limited to the presentation of the results, and a more in-depth literature review should be carried out and presented in the writing of the document.

The The introduction should include some more information on ecosystem services and the importance of trade-offs for land use and land management.

Figure 5 needs to be improved

139-143: need references

233. This section needs to be rewritten to improve understanding. Although all acronyms and abbreviations appear in the table, the paragraph needs to be understandable, not only with formulas and acronyms.

The discussion section is very brief. Since the results only deal with the raw data, the discussion section should make a deeper reflection of the results, adding the relevance of the results obtained for ecosystem-based management or the reflection on trade-offs beyond the data.

Reviewer 4 Report

I appreciate the quality of this research and its presentation.  For me, it would benefit from some additional description of the content of the Grain for Green Project, what its actions were. 

The manuscript should be checked for typos.  I noted some on lines 14,77,121,130 , 273, and 323, but there may be others.  In one case, WY and WC seem to be used interchangeably, which confused me because of the difference between water yield and water storage/conservation. 

I look forward to future publication that breaks down the watersheds into units that have had distinctive responses. There is broad description of these differences, but they are not as yet clear in the full presentation to provide sense of potential intervention strategy.

It is interesting that observation of Loess Plateau conditions by John Lowdermilk in the 1920's provoked his later establishment of the San Dimas Experimental Forest in California, after which he became the first director of the US Soil Conservation Service, established in the 1930's.  Now the research from LP provides valuable guidance for potential research in California.  I know that I want to share it with scientists in Northern California who are faced with similar challenges in identifying tradeoffs and synergies in restorative forestry.

Round 2

Reviewer 3 Report

The paper has been improved by the modifications made by the authors. I send one more comment to consider:

Line 302. A discussion sub-section has been created to talk about the limitations of the research, but only one sentence actually refers to limitations. The rest is repeated from the introduction. I suggest improving this subsection or incorporating it into another if it is too sparse.

Author Response

Dear Reviewers,

    Thank you for your careful comments and suggestions. The following answers are our revision and some thought according to your suggestions. Detailed revised portion are marked in red in revised manuscript.

Point : Line 302. A discussion sub-section has been created to talk about the limitations of the research, but only one sentence actually refers to limitations. The rest is repeated from the introduction. I suggest improving this subsection or incorporating it into another if it is too sparse.

Response : Thanks for your suggestions. We have adapted and expanded this section in the manuscript.

Special thanks to you for your good comments.

Best regards.

Yours sincerely,

Shutong Yang